# Insights into the Metabolite Differentiation Mechanism Between Chinese Dry-Cured Fatty Ham and Lean Ham Through UPLC-MS/MS-Based Untargeted Metabolomics

**DOI:** 10.3390/foods14030505

**Published:** 2025-02-05

**Authors:** Ruoyu Xie, Xiaoli Wu, Jun Hu, Wenxuan Chen, Ke Zhao, Huanhuan Li, Lihong Chen, Hongying Du, Yaqiong Liu, Jin Zhang

**Affiliations:** 1College of Food Science and Technology, Hebei Agricultural University, Baoding 071001, China; xieruoyu2023@126.com; 2State Key Laboratory for Quality and Safety of Agro-Products, Zhejiang Key Laboratory of Intelligent Food Logistic and Processing, Institute of Food Science, Zhejiang Academy of Agricultural Sciences, Hangzhou 310021, China; 18277970590@163.com (X.W.); hujun@zaas.ac.cn (J.H.); chenwx@zaas.ac.cn (W.C.); kzhao@snnu.edu.cn (K.Z.); huanhuanlee325@163.com (H.L.); cwc528@163.com (L.C.); 3Department of Food Science and Engineering, College of Light Industry and Food Engineering, Nanjing Forestry University, Nanjing 210037, China; hydu@njfu.edu.cn

**Keywords:** fatty ham, lean ham, Chinese dry-cured ham, untargeted metabolomics, UPLC-MS/MS, differential metabolite

## Abstract

To understand the impact and mechanism of removing fat and skin tissue on the nutritional metabolism of Chinese dry cured ham, the differential metabolites (DMs) profile between lean ham (LH) and fatty ham (FH) was explored though untargeted metabolomics based on UPLC-MS/MS. The results showed significant differences of the metabolite profiles between FH and LH. A total of 450 defined metabolites were detected, and 266 metabolites among them had significantly different abundances between the two hams, mainly including organic acids and derivatives, and lipids and lipid-like molecules, as well as organoheterocyclic compounds. Furthermore, 131 metabolites were identified as DMs, among which 101 and 30 DMs showed remarkably higher contents in FH and LH, respectively. The further Kyoto Encyclopedia of Genes and Genomes (KEGG) analysis suggested that DMs can be mostly enriched in the pathways of ABC transporters, amino acid biosynthesis, protein digestion and absorption, aminoacyl-tRNA biosynthesis, and 2-oxocarboxylic acid metabolism. Moreover, the metabolic network of DMs revealed that the prominent DMs in FH, such as 9(S)-HODE, 9,10-EpOME, 13-Oxo-ODE, L-palmitoyl carnitine, and D-fructose, were primarily involved in the endogenous oxidation and degradation of fat and glycogen. Nevertheless, the dominant DMs in LH, such as 2-isopropylmalic acid, indolelactic acid, and hydroxyisocaproic acid, were mainly the microbial metabolites of amino acids and derivates. These findings could help us understand how fat-deficiency affects the nutritional metabolism of Chinese dry-cured hams from a metabolic perspective.

## 1. Introduction

Chinese dry-cured ham is a traditional fermented meat product with a long history of over 1200 years, and its annual production has reached about 169,600 tons in 2023 [1,2]. Due to its bright color, pleasant flavor, rich nutrients, and good taste, the dry-cured ham is widely used as a flavor improver, umami enhancer, and nutrient fortifier in Chinese foods and dishes [2,3]. The production of Chinese dry-cured ham from fresh pig hind legs, including material screening, salting, dry-aging, and post-ripening processes, typically takes several months to ferment and convert food macromolecules into more absorbable nutrients [4]. Fats, proteins, and glycogens in raw hams degrade into large numbers of amino acids, fatty acids, and pyruvates during the dry-ripening process, which could further form some specific volatiles and metabolites via a variety of chemical reactions, contributing to the unique flavor and taste of Chinese dry-cured ham [5]. Therefore, fat plays an important role in the quality formation of Chinese dry-cured hams.

However, fat can also be a double-edged sword in the processing of Chinese dry-cured hams. On one hand, the high content of fats or polyunsaturated fatty acids is reported to be sensitive to oxidative deterioration, which may reduce the consumer’s overall acceptance of Chinese dry-cured hams [6]. On the other hand, the excessive consumption of fats, especially saturated fats, may contribute to an increased risk of cardiovascular diseases, such as obesity, arteriosclerosis, and coronary heart disease [7]. As a result, the dry-cured lean ham (LH), prepared from pig hind legs by removing skin and subcutaneous fat tissues, is becoming increasingly popular among consumers with health concerns. Zhang et al. [1] found that the fat-removal allowed Jinhua hams to produce more volatile branched alcohols and acids, mainly derived from the Strecker reaction or microbial metabolism of amino acids degraded from proteins, which significantly affected the flavor characteristics, color, and pH value of hams. Huang et al. [2] also reported that fat-deficiency could significantly promote the structure protein degradation, amino acid metabolism, and oxidative phosphorylation of Chinese dry-cured hams through a label-free proteomics strategy [1]. However, there is still a lack of a comprehensive understanding of the metabolites profile of the main nutrients in Chinese dry-cured LH. Moreover, how fat-deficiency affects the nutritional metabolism of Chinese dry-cured hams was also not fully investigated yet.

Metabolomics is an essential and scientific tool for identifying and analyzing metabolites with the advantages of a high sensitivity and high resolution [8,9]. In recent years, the technological advances in instruments, such as ultra-high-performance liquid chromatography (UPLC) and time-of-flight mass spectrometry (TOF-MS), have promoted the application of LC-MS-based metabolomics [10,11]. Therefore, the aim of the present study was (i) to investigate the differential metabolites (DMs) profile between Chinese dry-cured fatty ham (FH, without removing skin and subcutaneous fats) and LH using a UPLC-MS/MS-based untargeted metabolomics strategy, and (ii) to explore the potential mechanism of fat-deficiency influencing the metabolism of the main nutrients in Chinese dry-cured hams from the perspective of metabolite differentiation. The information obtained from the present study is helpful for understanding the effect of fat-deficiency on the nutritional metabolism of Chinese dry-cured hams.

## 2. Materials and Methods

### 2.1. Materials

The Chinese dry-cured hams were prepared and sampled in Jinnian Ham Co., Ltd. (Jinhua, China). After slaughtered and stored at 4 °C for 48 h, six trimmed pig hind legs with an average weight of 14.5 ± 0.5 kg (pH= 5.9 ± 0.2) were used for the preparation of Chinese dry-cured fatty hams (FH) and lean hams (LH). The ammonium acetate, ammonium hydroxide, methanol, and acetonitrile were bought from Sigma-Aldrich Co., Ltd. (St. Louis, MO, USA) and China National Medicines Co., Ltd. (Beijing, China).

### 2.2. Processing of FH and LH

The Chinese dry-cured hams were manufactured following the procedures of Zhou et al. [12] with some modifications. For the Chinese dry-cured FH, three whole hind legs were salted using 0.014% NaNO_2_ and 10% NaCl for 75 d, soaked and cleaned for 1 d, and then dried for 1 d. Subsequently, the hind legs were dehydrated for 7 d, sun-dried for 1 d, and then dry-ripened for 180 d. During the dry-ripening process, the humidity gradually decreased from 85% to 65% and the ambient temperature increased from 5 °C to 35 °C. After the dry-ripening, the legs were post-ripened at room temperature (25 °C) for about 1 month until their weights were reduced by 40% of the original weights. For the Chinese dry-cured LH, the skin and fatty tissues were trimmed from the other three hind legs, and the final hams were obtained following the same procedures as the dry-cured FH as mentioned above. In addition, for each dry-cured FH and LH, an internal sample was taken from the central part of the *biceps femoris* (about 3–4 cm depth). All samples were vacuum-filled and stored at −80 °C.

### 2.3. Extraction of Metabolites

The metabolites were extracted according to the method of Zhang et al. [13] with slight modifications. Firstly, samples were slowly thawed at 4 °C; then, approximately 20 mg sample was vortexed (MX-S, Scilogex, CT, USA) with cooled methanol/acetonitrile/water solution (2:2:1, *v*/*v*), and sonicated (KQ5200E, Kun Shan Ultrasonic Instruments Co., Ltd., Suzhou, China) at 60 Hz in an ice water bath for 30 min. After setting at −20 °C for 10 min, the sample mixture was centrifuged at 14,000× *g* for 20 min at 4 °C using a 5430R refrigerated microcentrifuge (Eppendorf^TM^, Hamburg, Germany). The supernatant was collected, vacuum-dried, and redissolved with the acetonitrile/water solution (1:1, *v*/*v*), followed by a vortex for 30 s and another centrifugation at 14,000× *g* for 15 min at 4 °C. Then, the final supernatant was collected and utilized for the UPLC-MS/MS-based metabolomics analysis.

### 2.4. UPLC-MS/MS Analysis

Based on the procedures of Zhang et al. and Wang et al. [11,13] with some modifications, the untargeted metabolomics analysis of ham samples was performed using an ultra-high-performance liquid chromatography (UPLC) system (1290 Infinity LC, Agilent Technologies Inc., Santa Clara, CA, USA) coupled to a triple time-of-flight mass spectrometer (TOF-MS) (6600, AB SCIEX, Foster City, CA, USA). Then, 2 µL samples were injected and separated through an ACQUIY UPLC BEH column (1.7 µm, 2.1 mm × 100 mm, Waters Inc., Wexford, Ireland) at 25 °C with the flow rate of 0.5 mL/min. The mobile phase contained 25 mM ammonium acetate and 25 mM ammonium hydroxide in water (A) and 100% acetonitrile (B). The chromatographic gradient was conducted with the following procedure: 0–0.5 min, kept at 95% B; 0.5–7 min, linearly reduced to 65% B; 7–8 min, further linearly reduced to 40% B; 8–9 min, kept at 40% B; 9–9.1 min, linearly increased to 95% B; and 9.1–12 min, kept at 95% B. Additionally, the quality control (QC) samples were also prepared and analyzed for the determination of stability and repeatability of instrumental analysis with the results shown as Appendix A.

The triple TOF-MS was operated at both positive (ESI+) and negative (ESI−) ion modes with the ESI source conditions of ion source Gas1 and Gas2 60, curtain gas 30, source temperature 600 °C, and ion spray voltage floating ± 5500 V. The MS-only acquisition was performed over 60–1000 Da (*m*/*z*) with the accumulation time for TOF MS scan of 0.20 s/spectra, while the auto MS/MS acquisition was conducted over 25–1000 Da (*m*/*z*) with the accumulation time for product ion scan at 0.05 s/spectra. The product ion scan was acquired under the conditions of information-dependent acquisition with the mode of high sensitivity, collision energy fixed at 35 V ± 15 eV, and declustering potential of ± 60 V, and excluded isotopes within 4 Da.

### 2.5. Metabolite Identification and Quantitative Analysis

The raw data from the MS/MS system were processed with ProteoWizard v3.0.4472 and XCMS v3.20.0 software for peak alignment, retention time correction, and peak area extraction under the following parameters: centWave *m*/*z*, 25 ppm; peak width, c (10, 60); prefilter, c (10, 100); bw, 5; mzwide, 0.025; and minfrac, 0.5. Meanwhile, the CAMERA (Collection of Algorithms of MEtabolite pRofile Annotation) was used for annotation of isotopes and adducts. Moreover, only the variables in the extracted ion features with >50% non-zero measurement values in at least one group were kept. Finally, metabolites were identified by comparing accurate *m*/*z* values (<25 ppm), retention times, MS/MS spectra, and collision cross-section values against an in-house database with available authentic standards, which was provided by Shanghai Applied Protein Technology Co., Ltd. (Shanghai, China).

The Pareto-scaled principal component analysis (PCA) and hierarchical cluster analysis (HCA) were performed to characterize the similarities and differences of metabolites between LH and FH. The orthogonal partial least-squares discriminant analysis (OPLS-DA) was used to identify the key metabolites in LH and FH. The robustness of the OPLS-DA model was also tested using 7-fold cross-validation and response permutation test (RPT), and the quality was evaluated by the parameters R^2^X, R^2^Y, and Q^2^. The variable influence in projection (VIP) value was further calculated from OPLS-DA to evaluate the contribution of metabolite to classification, and metabolites with both VIP > 1 and *p* < 0.05 were identified as statistically differential metabolites (DMs) [14].

### 2.6. Bioinformatic Analysis

The DMs were subjected to the bioinformatic analysis based on the online Kyoto Encyclopedia of Genes and Genomes (KEGG) database (http://geneontology.org/, accessed on 26 November 2020) to retrieve their KEGG orthology identifications and map to metabolic pathways. Subsequently, the enrichment analysis was performed using the Fisher’s exact test, considering the whole quantified metabolites as a background dataset. The Benjamini–Hochberg correction for multiple tests was further utilized for the adjustment of derived *p*-values. Only functional classifications and pathways with *p* < 0.05 were regarded as significant.

### 2.7. Statistical Analysis

All experiments were performed in biological pentaplicate. Figures were drawn with the Origin V2021 (OriginLab, Northampton, MA, USA) and Microsoft PowerPoint 2019 software. Analysis of variance (ANOVA) was conducted by the SPSS V27.0 (SPSS software Inc., Chicago, IL, USA). The PCA and OPLS-DA were conducted using the R package V4.0.2, while the HCA was performed with the plug-in applications in Origin V2021. The Waller–Duncan test was used to establish the difference among mean values, and the statistically significant difference was confirmed when *p* < 0.05.

## 3. Results and Discussion

### 3.1. Overall Difference in Metabolite Profiles

As an unsupervised learning method, PCA was used for the overall metabolite profile analysis of FH and LH groups, including both inter-group and intra-group variations. Moreover, as a supervised learning method with more focuses on inter-group variations, OPLS-DA was also used mainly for the metabolite differentiation analysis between FH and LH, especially for the DM screening based on VIP values in the further analysis. The PCA and OPLS-DA at both ESI+ and ESI− modes were carried out, and the results are shown in Figure 1. It is shown that, for metabolites detected at ESI+ (Figure 1a), 74.63% of the total variation can be explained by the first axis of PCA (PC1) and 83.28% by the first two axes (PC1 + PC2). Meanwhile, for metabolites detected at ESI− (Figure 1b), 71.15% of the total variation between FH and LH was attributed to PC1 from PCA and 79.70% to PC1 + PC2. These results indicated a statistical separation between the metabolite profiles of FH and LH whether at the ESI+ or ESI− modes. Furthermore, as presented in Figure 1c, 70.40% of the total variation can be explained by the first axis of OPLS-DA (PC1) and 79.89% by the first two axes (PC1 + PCo1) for metabolites found at ESI+. Moreover, as shown in Figure 1d, 69.80% of the total variation between metabolites from FH and LH found at ESI− can be attributed to PC1 and 78.98% to PC1 + PCo1 of OPLS-DA. These results are corresponding to the PCA results (Figure 1a,b), revealing that FH and LH had significantly distinctive metabolite compositions.

To evaluate the quality of OPLS-DA, the seven-fold cross-validation and RPT were also conducted with the results illustrated in Figure 1e,f. It is clear that, after the seven-fold cross-validation, the quality parameters of the OPLS-DA model were R^2^X = 0.599, R^2^Y = 1.000, and Q^2^ = 0.983 at ESI+ (Figure 1e), and R^2^X = 0.589, R^2^Y = 0.999, and Q^2^ = 0.979 at ESI− (Figure 1f). All these indicators were within a reasonable range (R^2^Y near to 1 and Q^2^ > 0.5) [15], indicating a good robustness and reliability of the OPLS-DA model whether at ESI+ or ESI−. Moreover, after the further 200-times RPT, the randomly permutated OPLS-DA model showed the quality parameters of R^2^ (X = 0) = 0.9346 and Q^2^ (X = 0) = −0.3375 at ESI+ (Figure 1e), as well as R^2^ (X = 0) = 0.8808 and Q^2^ (X = 0) = −0.4216 at ESI− (Figure 1f), suggesting that the original OPLS-DA model was not over-fitted at both ESI+ and ESI−.

### 3.2. Characterization of Metabolite Profiles

The volcano diagram was used for the visualization of the metabolite comparison between FH and LH, which was shown in Figure 2. As shown in Figure 2a, overall, 329 defined metabolites were identified at ESI+ from FH and LH, among which 174 metabolites were significantly up-regulated in FH (*p* < 0.05) and another 23 metabolites were remarkably up-regulated in LH (*p* < 0.05). Additionally, it is clear in Figure 2b that 121 defined metabolites were identified at ESI− from the two hams, with 40 and 29 among them showing significant up-regulation in FH and LH, respectively (*p* < 0.05). Therefore, overall, 450 defined metabolites were identified by UPLS-MS/MS-based untargeted metabolomics, among which 266 metabolites had significantly different abundances between FH and LH (*p* < 0.05). The identified information of these defined metabolites was shown in Appendix A.

These metabolites with different abundances were further classified according to the chemical taxonomy, and the results are exhibited in Figure 3. As illustrated in Figure 3a, organic acids and derivatives were the most numerous among them at the superclass level and accounted for 52.67%, followed by lipids and lipid-like molecules (17.78%) and organoheterocyclic compounds (10.00%). Furthermore, in terms of the class category, carboxylic acids and derivatives (48.89%) and fatty acyls (11.78%) were the main metabolites with different abundances (Figure 3b), which belonged to the organic acids and derivatives and lipids and lipid-like molecules, respectively. Moreover, when these metabolites were further classified into various subclasses, the predominant metabolites became amino acids, peptides, and analogues (47.78%), and fatty acids and conjugates (8.22%), as well as carbohydrates and carbohydrate conjugates (4.22%), which, overall, accounted for more than 60% of the overall metabolite species with different abundances. In addition, these three kinds of key metabolites are all metabolic products of major nutrients in dry-cured hams, including proteins, fats, and glycogens. Zhang et al. [1] reported that the discriminative flavor compounds between dry-cured FH and LH were mainly degradative, oxidative, or metabolic products of proteins and fats. Huang et al. [2] also found that the key differential proteins for the comparison of Jinhua FH vs. LH were mostly involved in the metabolic pathways of fats, glycogens, and proteins. These reports are both in accordance with our findings, revealing that the metabolites with different abundances were closely associated with their distinct mechanisms of nutritional metabolism between FH and LH.

### 3.3. Identification of DMs Between FH and LH

According to Zhang et al. [13], the metabolites with different abundances (*p* < 0.05) were further screened based on the VIP value from OPLS-DA. The metabolites with VIP > 1 among them were finally identified as DMs, which are shown in Figure 4 with their fold changes (FC) for the comparison of FH vs. LH. It is clear that 131 metabolites were identified as DMs, including 101 found at ESI+ (Figure 4a) and the other 30 determined at ESI− (Figure 4b). Li et al. [16] found 153 key DMs from Jinhua hams at different ripening stages, which were mostly similar with the DMs found in our study. Jiang et al. [17] reported the increased contents of nicotinic acid, betaine, and palmitoyl carnitine in dry-cured lamb sausages subjected to inoculated fermentation [17], which were also identified as the key DMs between FH and LH in the above findings. Moreover, most DMs (101 metabolites) were found to be significantly more abundant in FH (*p* < 0.05), among which 65 DMs showed a FC > 2, mainly including 40 dipeptides (Phe-Asn, Val-Tyr, etc.), 10 amino acids and analogues (1-aminocyclopropanecarboxylic acid, L-pyroglutamic acid, etc.), 8 fatty acids and derivatives (cis-9-palmitoleic acid, 13-Oxo-9E,11E-octadecadienoic acid (13-OxoODE), etc.), 4 nucleotides and derivatives (allopurinol riboside, inosine, etc.), and 3 carbohydrate and carbohydrate conjugates (glyceric acid, N-acetyl-D-lactosamine, and D-Tagatose) (Figure 4). In contrast, there were still 30 DMs possessing a significantly higher abundance in LH (*p* < 0.05), among which 12 were identified with a FC < 0.5, including Glu-Pro, 2-methylglutaric acid, 2-Isopropylmalic acid, adenine, L-iditol, thioetheramide-PC, indolelactic acid, palmitoyl ethanol amide, and some other compounds (Figure 4).

The HCA was further performed for the overall analysis of DMs identified at both ESI+ and ESI−, and the results were displayed as a tree-type heat map in Figure 5. On one hand, it is shown that 10 FH and LH samples were divided into two clusters. This result showed that pentaplicate samples from FH or LH can be clustered in the same subset (Euclidean distance < 4.0), suggesting that the difference in DMs in abundance can be contributed by the different processes between the two samples (with/without fat and skin tissues). On the other hand, 131 DMs can also be classified into two groups with the Euclidean distance < 3.0. One group included all DMs exhibiting a relatively higher abundance in LH (*p* < 0.05), and the other group was composed of all DMs that are significantly more abundant in FH (*p* < 0.05).

Noteworthily, dipeptides and free amino acids (FAAs) were the most numerous (>60% totally) among these dominant DMs in FH, accounting for approximately 53.47% and 7.92% of the total species, respectively (Figure 5). The low-molecular-weight peptide and FAA are mainly the degradation products of myofibrils [18] and considered two of the most important taste compounds in dry-cured meats [19]. Huang et al. [2] have reported that the myosin, the most important component of myofibrils, are more abundant in Jinhua LH than that in Jinhua FH. This finding implied that more structural proteins in FH were degraded than those in LH, which could be a main contributor of the higher dipeptide and FAA abundances in FH (Figure 4 and Figure 5), potentially leading to a richer taste of FH. Moreover, Zhang et al. [1] found that fat-removal allowed more FAAs in Jinhua hams to convert to certain volatile flavor compounds, mainly including branched alcohols and acids, under the metabolism of some microorganism genus, such as *Yamadazyma* and *Lactobacillus*. This can be another important reason for the relatively lower FAA abundance in LH (Figure 4 and Figure 5).

### 3.4. KEGG Pathway Enrichment Analysis

The KEGG pathway enrichment analysis can provide a better understanding of DM-related pathways [20]. Therefore, DMs from the FH vs. LH comparison were further analyzed, annotated, and attributed using the KEGG database with the results shown in Figure 6. As described in Figure 6b, the ABC transporters (ssc02010) and amino acid biosynthesis (ssc01230) pathways were associated with the most numerous DM species (11 and 10, respectively) among all the KEGG-enriched pathways, followed by the protein digestion and absorption (six DMs, ssc04974), aminoacyl-tRNA biosynthesis (six DMs, ssc00970), and 2-oxocarboxylic acid metabolism (six DMs, ssc01210) pathways. It is also presented in Figure 6a that these aforementioned pathways showed the highest significance level (log_10_
*P* < −4) except the 2-oxocarboxylic acid metabolism pathway (log_10_
*P* < −2). Moreover, as displayed in Figure 6c, the DMs can be mainly enriched to 20 metabolic pathways with |DAScore| ≥ 0.5, and half of these pathways can be attributed to metabolism. Noteworthily, the metabolism or biosynthesis of amino acids, including histidine, β-alanine, glycine, serine, threonine, alanine, aspartate, glutamate, and arginine (ssc00340, ssc00410, ssc00260, ssc00250, and ssc00220), accounted for five pathways among the ten metabolism-associated pathways. Huang et al. [2] reported that the differential proteins between Jinhua LH and FH can be mainly attributed to the metabolism of amino acids, carbohydrate, and energy after the KEGG pathway analysis, which was partially in consistent with our above findings.

### 3.5. Connections Between DMs and Metabolic Pathway

Based on the DM profiles and KEGG enrichment analysis, the potential metabolic network of DM formation and alteration from FH and LH is illustrated in Figure 7. It is clear that 14 key metabolic pathways were involved in this DM metabolic network, such as glycerophospholipid metabolism, glycine, serine, and threonine metabolism, alanine, aspartate, and glutamate metabolism, β-alanine metabolism, glutathione metabolism, linoleic acid metabolism, and phenylalanine metabolism. Meanwhile, all these key pathways were found closely associated with the metabolisms of three major nutrients in hams, glycogen, protein, and fat (Figure 7). Briefly, the absence of fats significantly affected the nutrient metabolism process and taste-related compound formation in LH. On one hand, the relatively higher contents of 2-isopropylmalic acid, indolelactic acid, and hydroxyisocaproic acid in LH (Figure 4b, *p* < 0.05) revealed its more active microbial metabolism of amino acids. Hydroxyisocaproic acid can be produced by some microorganisms, such as lactic acid bacteria and *Clostridium*, through leucine degradation [21]. Some leucine-requiring yeasts were probably responsible for the accumulation of 2-isopropylmalic acid in dry-cured hams [22]. Meanwhile, tryptophan can be converted to indolelactic acid and derivatives under the involvement of *Lactobacillus* [23]. Moreover, the remarkably more abundant N-acetyl-L-phenylalanine and DL-3-phenyllactic acid in LH (Figure 4b, *p* < 0.05) may also suggest its higher activity of microbial metabolism of phenylalanine [24,25]. Corresponding with this finding, Huang et al. [2] also found that some proteins associated with phenylalanine metabolism and synthesis were significantly up-regulated in fat-deficient Jinhua hams.

On the other hand, the lack of fat tissues in LH contributed to an overall down-regulated fat metabolism, resulting in the relatively lower abundances of 9S-hydroxyoctadeca-10,12-dienoic acid (9(S)-HODE), 9,10-Epoxyoctadecenoic acid (9,10-EpOME), and 13-Oxoctadecadienoic acid (13-Oxo-ODE) (Figure 4b, *p* < 0.05), all of which were metabolic products of linoleic acid, an important polyunsaturated fatty acid in human nutrition and diet [26]. It has been reported that linoleic acid can be oxidized to hydroperoxylinoleic acid (HPODE), an important precursor of 9(S)-HODE, in the presence of lipoxygenase [27]. 9,10-EpOME was also reported to be produced from linoleic acid under the epoxidation role of cytochrome P450 cyclooxygenase [28]. Oxo-ODE, as a peroxisome proliferator-activated receptor agonist, is also produced from the lipoxygenase/linoleate system [29]. Meanwhile, previous studies showed that HODEs are widely present in foods rich in fats or oils with high-temperature or long-time processing, particularly abundant in cured fatty meats [27,30], which was consistent with our above findings. Moreover, L-palmitoyl carnitine was also remarkably down-regulated in LH compared with FH (Figure 4a, *p* < 0.05), which is mainly involved in the β-oxidation of fatty acids [31]. Furthermore, betaine, playing an important role in glutathione metabolism and the conversion of glycine to serine, is also the oxidation product of choline and the methylation product of glycine [32,33]. This can be responsible for the significant up-regulations of glutathione metabolism and glycine, serine, and threonine metabolism in FH (Figure 6c, *p* < 0.05), as well as the relatively lower abundances of betaine and choline in LH (Figure 4a, *p* < 0.05). In addition, some glycolysis-related metabolites, such as D-fructose and D-tagatose, were found with significantly lower contents in LH (Figure 4b, *p* < 0.05), implying a probable down-regulated glycogen metabolism compared with FH. Huang et al. [2] have reported the lower level of glycolysis/gluconeogenesis in fat-removal Jinhua LH, resulting from some down-regulated enzymes compared with FH, such as glucose-6-phosphate isomerase, fructose-bisphosphatase, and ATP-dependent 6-phosphofructokinase. Bautista et al. [34] also reported the negative correlation between the D-tagatose and growth of some spoilage-causing bacteria. However, the mechanisms of their differentiation in FH and LH are still not fully understood and require further investigation.

In summary, the dominant DMs in FH mainly focused on the endogenously oxidative and degraded metabolites of fat and glycogen, while the prominent DMs in LH were primarily the microbial metabolites of amino acids and derivates. Zhang et al. [1] found that the discriminative flavor compounds of Jinhua normal hams were mostly β-oxidation and the degradation products of fatty acids, whereas those of fat-removal hams were mainly derived from the Strecker reaction or microbial metabolism of amino acids. Huang et al. [2] reported that the dominant discriminative proteins from Jinhua fatty hams were mainly involved in glycogen metabolism, while those from fat-deficient hams mostly participated in the amino acid metabolism and oxidative phosphorylation. These reports were both partially consistent with our findings.

## 4. Conclusions

In this study, the metabolite differentiation mechanism between Chinese dry-cured FH and LH was explored through untargeted metabolomics based on UPLC-MS/MS. The PCA and OPLS-DA results showed a significant difference in metabolite profiles for FH and LH. Specifically, overall, 450 defined metabolites were detected, among which 266 metabolites had significantly different abundances between the two hams, mainly including organic acids and derivatives (52.67%), lipids and lipid-like molecules (17.78%), and organoheterocyclic compounds (10.00%). Then, 131 metabolites were further identified as DMs based on OPLS-DA (VIP > 1 and *p* < 0.05), which included 101 and 30 DMs remarkably more abundant in FH and LH, respectively. Furthermore, the KEGG analysis revealed that most DMs can be enriched in the pathways of ABC transporters, amino acid biosynthesis, protein digestion and absorption, aminoacyl-tRNA biosynthesis, and 2-oxocarboxylic acid metabolism. Moreover, the metabolic network showed that the dominant DMs in FH, such as 9(S)-HODE, 9,10-EpOME, 13-Oxo-ODE, L-palmitoyl carnitine, and D-fructose, were mainly the endogenously oxidative and degraded metabolites of fat and glycogen. However, the prominent DMs in LH, such as 2-isopropylmalic acid, indolelactic acid, and hydroxyisocaproic acid, were primarily associated with the microbial metabolism of amino acids and derivates. The information obtained from this study could help understand how fat-deficiency affects the nutritional metabolism of Chinese dry-cured hams from a metabolic perspective. However, further investigation is still required for the molecular and microbiological mechanism elucidation of metabolite differentiation derived from fat-deficiency.

## Figures and Tables

**Figure 1 foods-14-00505-f001:**
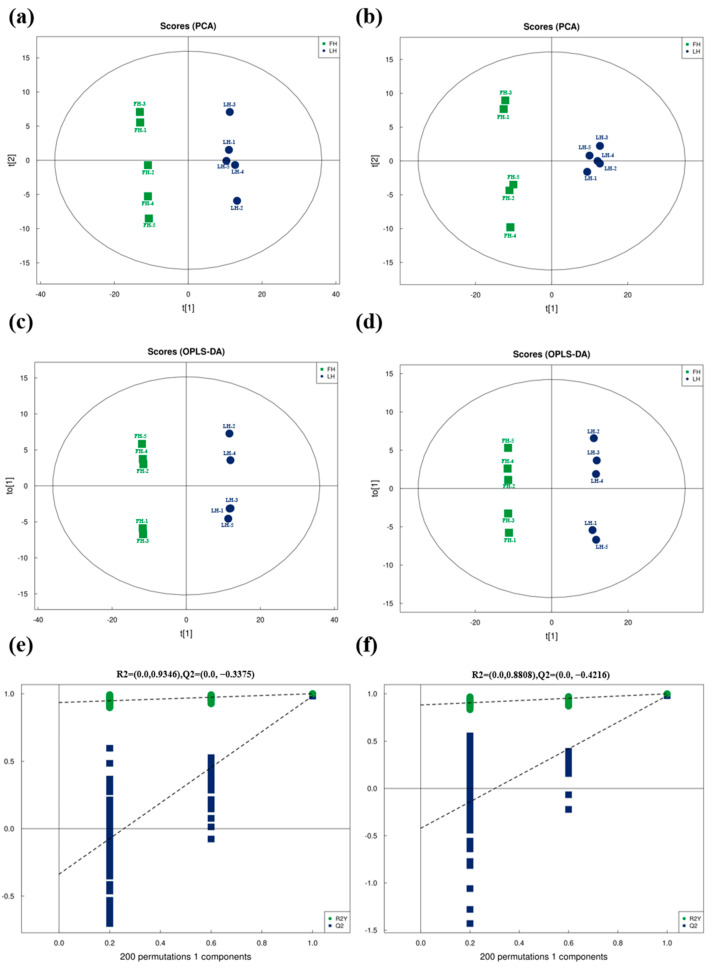
Principal component analysis (PCA) (**a**,**b**), and orthogonal partial least-squares discriminant analysis (OPLS-DA) (**c**,**d**), as well as 7-fold cross-validation and response permutation test (RPT) for OPLS-DA (**d**–**f**) of metabolites from FH and LH: (**a**,**c**,**e**) positive ion mode (ESI+); (**b**,**d**,**f**) negative ion mode (ESI−); FH, fatty ham; LH, lean ham.

**Figure 2 foods-14-00505-f002:**
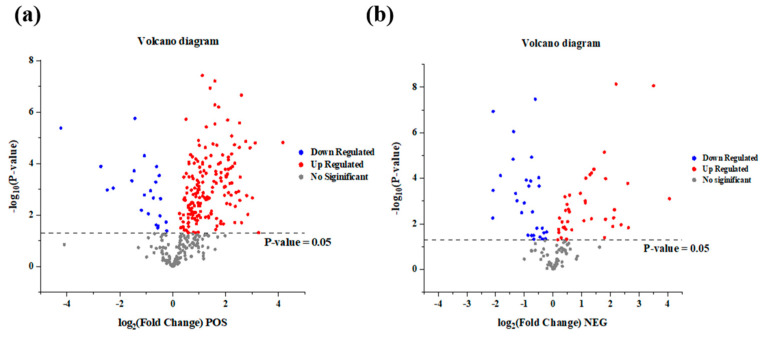
Volcano diagram of identified metabolites for the comparison of FH vs. LH at ESI+ (**a**) and ESI− (**b**). FH, fatty ham; LH, lean ham.

**Figure 3 foods-14-00505-f003:**
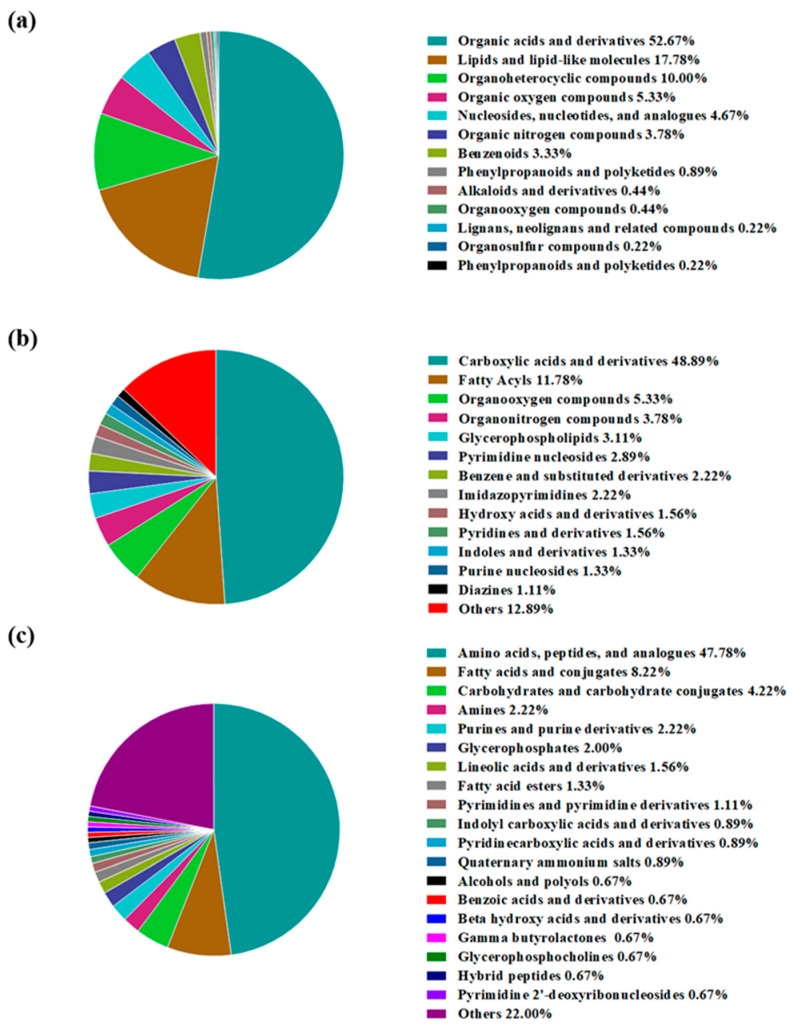
Classification of identified metabolites from FH and LH with significantly different abundances at superclass (**a**), class (**b**), and subclass (**c**) categories. FH, fatty ham; LH, lean ham.

**Figure 4 foods-14-00505-f004:**
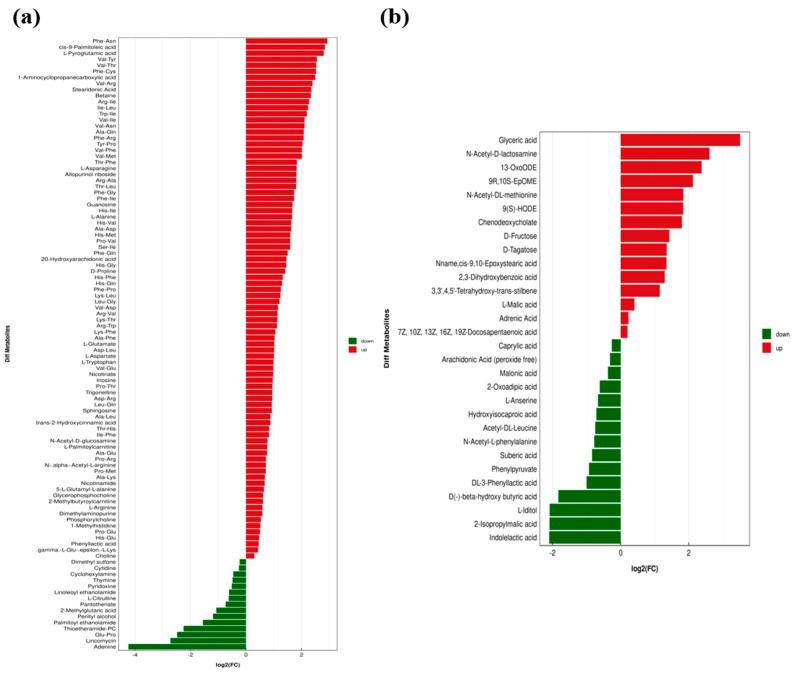
Fold changes (FC) of identified differential metabolites (DMs) from the comparison of FH vs. LH: (**a**) ESI+; and (**b**) ESI−. FH, fatty ham; LH, lean ham.

**Figure 5 foods-14-00505-f005:**
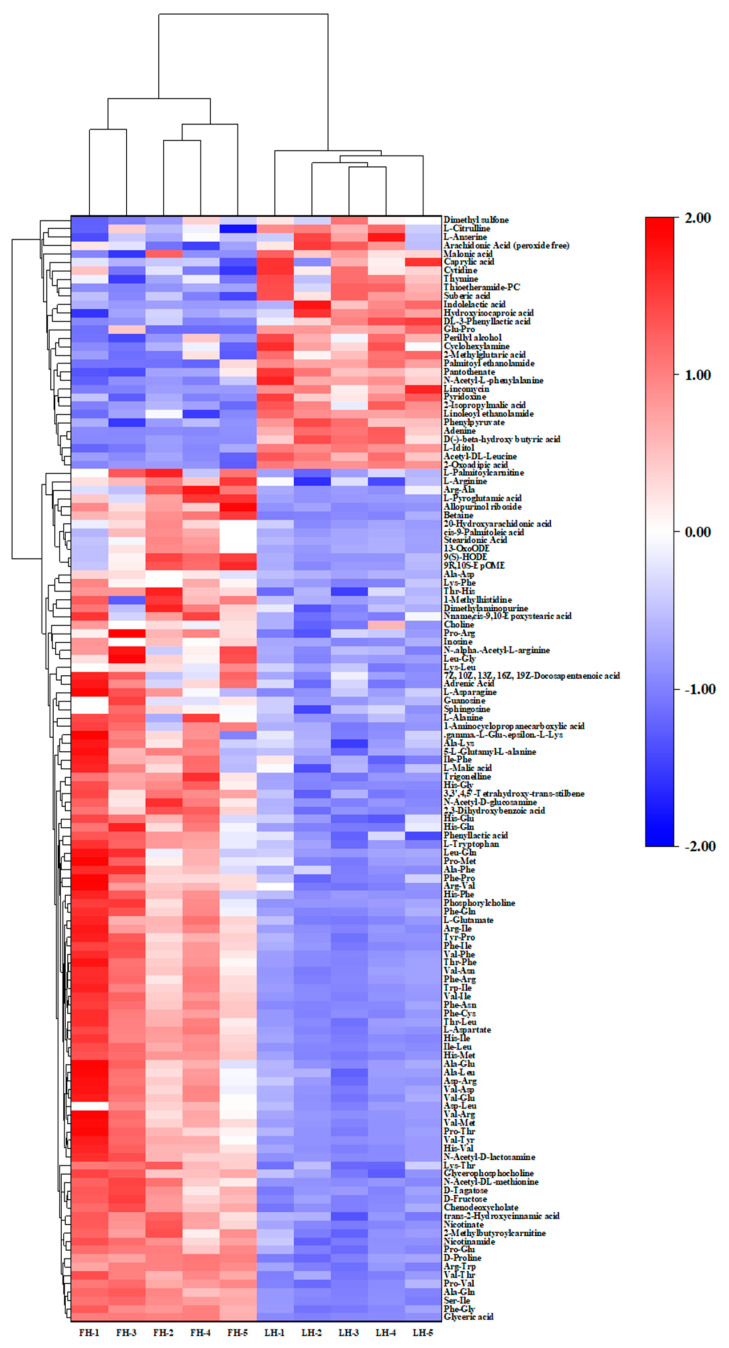
Hierarchical cluster analysis (HCA) of DMs from FH vs. LH comparison. The color gradation represents the Z-score of metabolite abundance. FH, fatty ham; LH, lean ham.

**Figure 6 foods-14-00505-f006:**
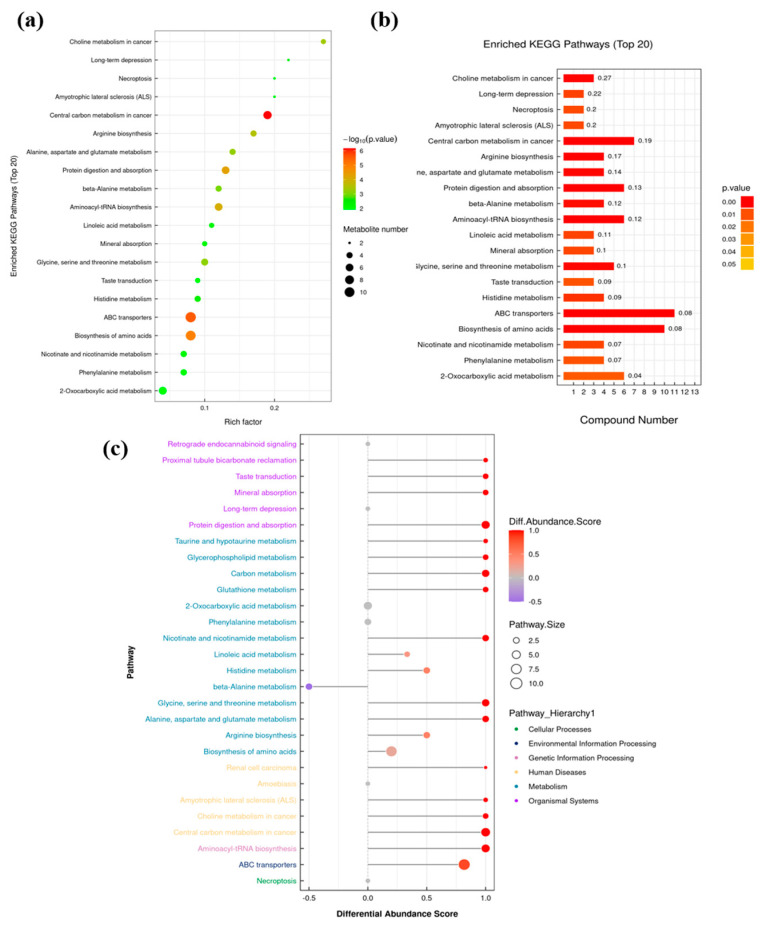
Kyoto Encyclopedia of Genes and Genomes (KEGG) pathway enrichment analysis of DMs from FH vs. LH comparison: (**a**) functional enrichment with rich factors; (**b**) annotation with metabolite numbers; and (**c**) attribution with differential abundance scores (DAScores). FH, fatty ham; LH, lean ham.

**Figure 7 foods-14-00505-f007:**
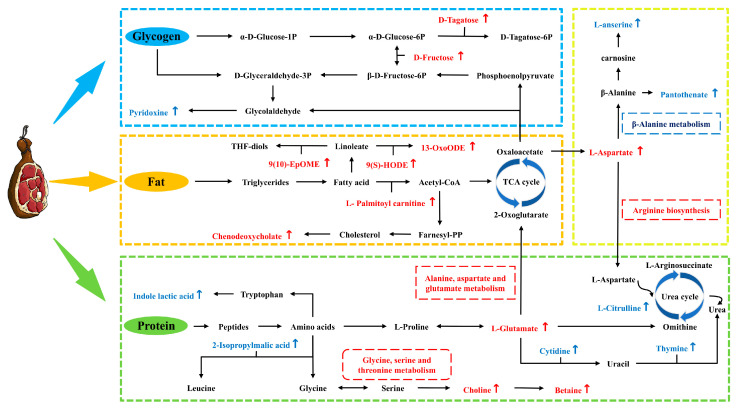
Potential metabolic pathways of key DMs from FH vs. LH comparison. The red and blue colors represent up-regulations of DMs and metabolic pathways in FH and LH, respectively. FH, fatty ham; LH, lean ham.

## Data Availability

The original contributions presented in the study are included in the article/Appendix A, further inquiries can be directed to the corresponding authors.

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
