# Peer review of "Insights into the Metabolite Differentiation Mechanism Between Chinese Dry-Cured Fatty Ham and Lean Ham Through UPLC-MS/MS-Based Untargeted Metabolomics"

_foods, 2025, doi:10.3390/foods14030505_

Round 1

Reviewer 1 Report

Comments and Suggestions for Authors

Abstract:

·      Line 24: “The further KEGG analysis suggested that DMs can be mostly enriched to the pathways associated with nutrient metabolism and organismal system, such as ABC transporters, amino acid biosynthesis, central carbon metabolism in cancer, protein digestion and absorption, aminoacyl-tRNA biosynthesis, and 2-oxocarboxylic acid metabolism.” :

1)        What is the ‘organismal system’? Is this correct? 

2)        How can the ‘carbon metabolism in cancer’ be affected in ham? 

Introduction:

·      Line 69: please alter, since UPLC-MS based metabolomics cannot be considered high-throughput. 

M&M:

·      Line 132: did the auto MS/MS acquisition entail a certain threshold? Information dependent acquisition is mentioned, but it is not clear how and when MS/MS was triggered exactly.

·      Section 2.5: Please include thresholds used for the validation of the OPLS-DA models and selection of DMs.

·      Line 145: “accuracy m/z values (<25 ppm)” However, 25 ppm is not considered highly accurate. High resolution MS usually entails < 5, max. 10 ppm?  

Results & Discussion:

·      Lines 175-188: The purpose of PCA and OPLS-DA is different, but the authors do not apply this. The use and purpose of comparing the two is not entirely clear here. 

·      Did the authors assess clustering of QC samples and signal drift? Can this be discussed, and maybe also visualized?

·      Line 206: e.g. 239 metabolites at ESI+ are mentioned, but are these all metabolites in ESI+, or the selected DMs only. Please clarify for all.

·      Section 3.3: Please clarify the level of identification of all metabolites. It was not entirely clear whether this was done based on KEGG match solely or by comparison to in-house standards here. Can a list of in-house standards please be provided (in supplementary for example)? In case identification is based on m/z match solely, and not by means of RT and MS/MS spectrum matching with analytical standards, the phrasing ‘putative’ or ‘tentative’ identification should be used (throughout).

·      Section 3.4: same remark concerning the “central carbon metabolism in cancer” as previously.

·      How do the authors explain the (merely) partially consistent findings or inconsistent findings? This is not sufficiently addressed.

·      Can the authors please discuss study strengths and weaknesses?

·      In the introduction it is mentioned that “The information obtained from the present study is helpful for the development of healthier fat-reduced dry-cured meats” This is however not addressed in the discussion, which is mainly focused on describing the results and how metabolites are different, but not what this entails in the context of the final consumer product, i.e. how the current work improves this understanding. 

Comments on the Quality of English Language

All is understandable, but improvements in language can be made. 

Author Response

Comment 1: Line 24: What is the ‘organismal system’? Is this correct?

Response 1: Thank you for your careful review and valuable comment. The “organismal systems” is an integral concept of organisms and a cluster of KEGG-enriched pathways. As illustrated in Figure 6c, it mainly includes protein absorption and digestion, mineral absorption, taste transduction, etc. We agree that the “organismal systems” is a too obscure concept to describe the KEGG pathway-enrichment profile. According to your valuable comment, the “organismal systems” was removed and this sentence has been changed to “The further Kyoto Encyclopedia of Genes and Genomes (KEGG) analysis suggested that DMs can be mostly enriched to the pathways of ABC transporters, amino acid biosynthesis, protein di-gestion and absorption, aminoacyl-tRNA biosynthesis, and 2-oxocarboxylic acid metabolism” in the revised manuscript (Page 1, Line 25-28).

Comment 2: Line 24: How can the ‘carbon metabolism in cancer’ be affected in ham?

Response 2: Thank you for your careful review and valuable comment. We agree that the “carbon metabolism in cancer” cannot be affected in hams. According to your valuable comment, the “carbon metabolism in cancer” was removed and this sentence has been changed to “The further Kyoto Encyclopedia of Genes and Genomes (KEGG) analysis suggested that DMs can be mostly enriched to the pathways of ABC transporters, amino acid biosynthesis, protein di-gestion and absorption, aminoacyl-tRNA biosynthesis, and 2-oxocarboxylic acid metabolism” in the revised manuscript (Page 1, Line 25-28).

Comment 3: Line 69: please alter, since UPLC-MS-based metabolomics cannot be considered high-throughput.

Response 3: Thank you for your valuable comment. We agree that the UPLC-MS based metabolomics has not been considered high-throughput already. According to your comment, the “high-throughput” has been removed in the revised manuscript (Page 2, Line 69-70).

Comment 4: Line 132: did the auto MS/MS acquisition entail a certain threshold? Information dependent acquisition is mentioned, but it is not clear how and when MS/MS was triggered exactly.

Response 4: Thank you for your careful review and valuable comment.

Firstly, there was not particular threshold used for our auto MS/MS acquisition, but a high sensitivity mode was used during the performance, which means the peak signal outputs with a relatively high response intensity were chosen for the further acquisition. Hence, the information “with the mode of high sensitivity” has been added in the revised manuscript according to your comment (Page 3, Line 135-137).

Secondly, some previously studies [1,2] also showed the same performance during the auto MS/MS acquisition of metabolomic profiling. They didn’t provide particular threshold either, but also adopted the high sensitivity mode. Therefore, we believe that this performance/condition is reasonable for our study.

References:

[1] Wang, J.; Xu, Z.; Zhang, H.; Wang, Y.; Liu, X.; Wang, Q.; Xue, J.; Zhao, Y.; Yang, S. Meat differentiation between pasturefed and concentrate-fed sheep/goats by liquid chromatography quadrupole time-of-flight mass spectrometry combined with metabolomic and lipidomic profiling. Meat Sci. 2021, 173, 108374.

[2] Zhang, J.; Li, X.; Zhao, K.; Li, H.; Liu, J.; Da, S.; Ciren, D.; Tang, H. In vitro digestion and fermentation combined with microbiomics and metabolomics reveal the mechanism of superfine yak bone powder regulating lipid metabolism by altering human gut microbiota. Food Chem. 2023, 410, 135441. 

Comment 5: Section 2.5: Please include thresholds used for the validation of the OPLS-DA models and selection of DMs.

Response 5: Thank you for your valuable suggestion. Actually, the thresholds for validating OPLS-DA model and DMs selection have already stated in the original manuscript. Metabolites with both VIP > 1 and p < 0.05 were identified as statistically DMs (Page 4, Line 154-157), and the OPLS-DA model with both R2Y near to 1 and Q2 > 0.5 was considered validated (Page 4, Line 199). 

Comment 6: Line 145: “accuracy m/z values (<25 ppm)” However, 25 ppm is not considered highly accurate. High resolution MS usually entails < 5, max. 10 ppm?

Response 6: Thank you for your valuable comments. We found that the tolerance range of <25 ppm is widely used in many previous studies associated with untargeted metabolomics investigations [1-4]. Therefore, we believe that it is accurate enough for our study even though <25 ppm maybe not regarded as high-accuracy. In addition, the “accuracy m/z values” has been corrected to “accurate m/z values” in the revised manuscript (Page 3, Line 146-148).

References:

[1] Chen, J.; Wang, W.; Kong, J.; Yue, Y.; Dong, Y.; Zhang, J.; Liu, L. Application of UHPLC-Q-TOF MS based untargeted metabolomics reveals variation and correlation amongst different tissues of Eucommia ulmoides Oliver. Microchem. J. 2022, 172, 106919. 10.1016/j.microc.2021.106919.

[2] Xu, J.; Su, G.; Huang, X.; Chang, R.; Chen, Z.; Ye, Z.; Cao, Q.; Kijlstra, A.; Yang, P. Metabolomic Analysis of Aqueous Humor Identifies Aberrant Amino Acid and Fatty Acid Metabolism in Vogt-Koyanagi-Harada and Behcet’s Disease. Front. Immunol. 2021, 12, 587393. 10.3389/fimmu.2021.587393.

[3] Yong, K.; Luo, Z.; Luo, Q.; Yang, Q.; Huang, Y.; Zhao, X.; Zhang, Y.; Cao, S. Plasma metabolome alteration in dairy cows with left displaced abomasum before and after surgical correction. J. Dairy Sci. 2021, 104, 8177. 10.3168/jds.2020-19761.

[4] Gao, Y.; Li, J.; Li, X.; Li, X.; Yang, S.; Chen, N.; Li, L.; Zhang, L. Tetrahydroxy stilbene glycoside attenuates acetaminophen‐induced hepatotoxicity by UHPLC‐Q‐TOF/MS‐based metabolomics and multivariate data analysis. J. Cell. Physiol. 2020, 236, 3832-3862. 10.1002/jcp.30127.

Comment 7: Lines 175-188: The purpose of PCA and OPLS-DA is different, but the authors do not apply this. The use and purpose of comparing the two is not entirely clear here.

Response 7: Thanks for your careful review and valuable comments. It is known that PCA is an unsupervised learning method, but OPLS-DA is a supervised learning method. Meanwhile, PCA is widely used in the overall analysis of metabolite profiles, whereas OPLS-DA is usually used in the classification and discriminant analysis of metabolites. Furthermore, PCA included both inter-group and intra-group variations, but OPLS-DA is more focused on inter-group variations rather than intra-group variations. The above statements (use, purpose, and comparison of PCA and OPLS-DA) have been added accordingly in the revised manuscript (Page 4, Line 176-181).

Comment 8: Did the authors assess clustering of QC samples and signal drift? Can this be discussed, and maybe also visualized?

Response 8: Thanks for your valuable comments. As we stated in the manuscript (Page 3, Line 127-129), the QC samples were also prepared and analyzed for the determination of stability and repeatability of instrumental analysis. According to your comment, the total ion chromatograms of QC samples at both ESI+ and ESI- have been added as Figure S1 in the Supplementary Materials of the revised manuscript (Page 13, Line 411-413). As shown in Figure S1, the total ion chromatograms of different QC samples at both ESI+ and ESI- were highly overlapped and rarely signal drifted, indicating good stability and repeatability of the UPLC-based analysis.

 Comment 9: Line 206: e.g. 329 metabolites at ESI+ are mentioned, but are these all metabolites in ESI+, or the selected DMs only. Please clarify for all.

Response 9: Thank you for your careful review and valuable suggestion. 329 was the total amount of metabolites identified at ESI+ for FH and LH excluding undefined compounds. According to your valuable suggestion, the “329 metabolites” has been modified to “totally 329 defined metabolites” in the revised manuscript (Page 6, Line 212-213). The same modifications were also made across the other places of the revised manuscript (Page 1, Line 20-21; Page 6, Line 215-216; Page 12, Line 393).

Comment 10: Section 3.3: Please clarify the level of identification of all metabolites. It was not entirely clear whether this was done based on KEGG match solely or by comparison to in-house standards here. Can a list of in-house standards please be provided (in supplementary for example)? In case identification is based on m/z match solely, and not by means of RT and MS/MS spectrum matching with analytical standards, the phrasing ‘putative’ or ‘tentative’ identification should be used (throughout).

Response 10: Thank you for your valuable comments. As we have stated in the Section 2.5. “Metabolite Identification and Quantitative Analysis” of the manuscript, all defined metabolites were identified by comparing accurate m/z values and MS/MS spectra against an in-house database with available authentic standards (Page 3, Line 146-148). During this process, the means of RT were also an important parameter to identify metabolites, but the KEGG match was not used. Therefore, we think the use of “identification” rather than “putative/tentative” was correct. 

Comment 11: Section 3.4: same remark concerning the “central carbon metabolism in cancer” as previously.

Response 11: Thank you for your careful review and valuable comment. We agree that the “carbon metabolism in cancer” cannot be affected in hams. According to your valuable comment, the “carbon metabolism in cancer” has been removed in the revised manuscript (Page 10, Line 305). 

Comment 12: How do the authors explain the (merely) partially consistent findings or inconsistent findings? This is not sufficiently addressed.

Response 12: Thank you for your valuable comments. Actually, as we discussed in the manuscript, most of our findings were consistent or largely consistent with previous reports, mainly including the findings of DMs associated with protein and fat metabolism. However, as we also stated in the manuscript (Page 11, Line 365-367), the results of some glycogen-associated DMs, such as D-fructose and D-tagatose, were not much consistent with previous reports. Although some explanations were given based on previously peer-reviewed studies, we think the mechanisms of their differentiation in FH and LH are still not fully understood and require further investigation, which have been addressed already in the manuscript (Page 11, Line 368-374). 

Comment 13: Can the authors please discuss study strengths and weaknesses?

Response 13: Thank you for your valuable suggestions and comments. On one hand, we believe the best strength of this study was that it provided a metabolic perspective for us to understand how fat-deficiency affects nutritional metabolism of Chinese dry-cured hams. This has been addressed in both Abstract and Conclusions sections of the revised manuscript (Page 1, Line 32-34; Page 12, Line 406-408). On the other hand, the quality formation of Chinese dry-cured hams is a complex process, including chemical reactions, physical changes, and microbiological fermentation. Therefore, other perspectives, such as enzyme activity changes and microbial evolution, are also important and necessary for the further investigation of molecular and microbiological mechanisms of metabolic differences. This significant weakness has also been added as “However, further investigation is still required for the molecular and microbiological mechanism elucidation of metabolite differentiation derived from fat-deficiency” in the Conclusions section of the revised manuscript (Page 12, Line 408-410). 

Comment 14: In the introduction it is mentioned that “the information obtained from the present study is helpful for the development of healthier fat-reduced dry-cured meats”. This is however not addressed in the discussion, which is mainly focused on describing the results and how metabolites are different, but not what this entails in the context of the final consumer product, i.e. how the current work improves this understanding.

Response 14: Thank you for your valuable comments. We agree that this research is mainly focused on the main differential mechanism of nutritional metabolisms between Chinese dry-cured FH and LH. According to your valuable comments, this sentence has been changes to “The information obtained from the present study is helpful to understand the effect of fat-deficiency on the nutritional metabolism of Chinese dry-cured hams” in the revised manuscript (Page 2, Line 78-80). Meanwhile, the related statements have also been modified in the Abstract (Page 1, Line 32-34) and Conclusions sections (Page 12, Line 406-408) of the revised manuscript. 

Comment 15: All is understandable, but improvements in language can be made.

Response 15: Thank you for your recognition and valuable suggestion. According to your suggestion, we have invited Professor Fanbin Kong (Department of Food Science and Technology, the University of Georgia, USA) to help us with the proof-reading of this manuscript. He is an expert in food science research and worked in the USA as a good English speaker for over 30 years. Our thanks to him for his help with language were also stated in the Acknowledgments section of the revised manuscript (Page 13, Line 424-426).

Reviewer 2 Report

Comments and Suggestions for Authors

Dear authors, the reviewed article is interesting.  

This work aimed to investigate the differential metabolite profiles among Chinese dry-cured hams with different fat contents by UPLC-MS/MS-based untargeted metabolomics and explore how fat deficiency affects the metabolism of key nutrients, contributing to the development of healthier and low-fat cured meat products.  

During this review, some minor points were detected that need to be modified in the text of the document: 

Line 22: use and instead of &

Line 25: What is the meaning of KEGG?

Line 40: Could you include information on this product's total production and per capita consumption?

Line 54: Do these diseases only develop from consuming large quantities, or is lifestyle also associated?

Line 58: Avoid using authors' names or surnames in the document. Only the reference number corresponding to the information is required, and it must be placed in brackets.

Line 62: Idem

Line 73: the differential metabolites (DMs)… like in the abstract

Line 83: The legs were used immediately after the animal was slaughtered, or the carcasses were stored for 24 h to take the piece later.

Line 90: Idem

Line 104: Idem

Line 105: Briefly, samples

Line 105: insert equipment information (model, brand, country)

Line 105: insert mixing conditions

Line 106: insert equipment information (model, brand, country)

Line 108: 14,000

Line 111: 14,000

Line 113: Analysis

Line 114: Idem

Line 120: °C

Line 179: 83.28%

Line 219: acids and derivatives

Line 220: lipids and lipid-like

Line 222: acids and derivatives

Line 223,224,226: use and instead of &

Line 230: Idem

Line 232: Idem

Line 241,246,248,284,309: Idem

Line 312: The resolution of the figure needs to be improved; when zooming in, the letter becomes distorted and it is difficult to read the text.

Line 321,322: use and instead of & (make change through document text)

Line 336: Idem

Line 351: remove text space… [27,30],

Line 364,370: Idem

Author Response

Comment 1: Line 22: use “and” instead of “&”.

Response 1: Thank you for your careful review and valuable suggestion. It has been changed in the revised manuscript accordingly (Pages 1, Lines 22-23).

Comment 2: Line 25: What is the meaning of KEGG?

Response 2: Thank you for your valuable comment. According to your comment, the full name of KEGG, Kyoto Encyclopedia of Genes and Genomes, has been added in the revised manuscript (Page 1, Line 25).

Comment 3: Line 40: Could you include information on this product's total production and per capita consumption?

Response 3: Thank you for your valuable comment. The total production of Chinese dry-cured hams has been reported to reach about 169,600 tons in 2023 [1,2]. However, we didn’t find the data about per capita consumption of Chinese dry-cured hams. Therefore, the information of the annual production of Chinese dry-cured hams has been added in the revised manuscript (Page 1, Line 40).

References:

[1] Zhang, J.; Zhao, K.; Li, H.; Li, S.; Xu, W.; Chen, L.; Xie, J.; Tang, H. Physicochemical property, volatile flavor quality, and microbial community composition of Jinhua fatty ham and lean ham: A comparative study. Front. Microbiol. 2023, 14, 1124770.

[2] Huang, Q.; Xie, R.; Wu, X.; Zhao, K.; Li, H.; Tang, H.; Du, H.; Peng, X.; Chen, L.; Zhang, J. Insights into the protein differentiation mechanism between Jinhua fatty ham and lean ham through Label-Free Proteomics. Foods 2023, 12, 4348. 

Comment 4: Line 54: Do these diseases only develop from consuming large quantities, or is lifestyle also associated?

Response 4: Thank you for your careful review and valuable comment. To our knowledge, cardiovascular disease is a complex disease caused by a combination of factors, such as poor lifestyle, genetic factors, environmental factors, and psychological factors. According to your comment, the “could contribute to” has been corrected to “may contribute to” in the revised manuscript (Page 2, Line 55-57). 

Comment 5: Line 58: Avoid using authors' names or surnames in the document. Only the reference number corresponding to the information is required, and it must be placed in brackets.

Response 5: Thank you for your careful review. To our knowledge, this kind of citation meets the requirement of this journal. The same type of citation can also be found in the articles published on this journal, such as Huang et al. [1] and Xu et al. [2].

References:

[1] Huang, Q.; Xie, R.; Wu, X.; Zhao, K.; Li, H.; Tang, H.; Du, H.; Peng, X.; Chen, L.; Zhang, J. Insights into the protein differentiation mechanism between Jinhua fatty ham and lean ham through Label-Free Proteomics. Foods 2023, 12, 4348.

[2] Xu, J.; Liu, Y.; Ma, J.; Li, P.; Geng, Z.; Wang, D.; Zhang, M.; Xu, W. Recombinant porcine 12-lipoxygenase catalytic domain: Effect of inhibitors, selectivity of substrates and specificity of oxidation products of linoleic acid. Foods 2022, 11, 980.

Comment 6: Line 62: Idem

Response 6: The same to Response 5.

Comment 7: Line 73: the differential metabolites (DMs)… like in the abstract.

Response 7: Thank you for your valuable suggestion. It has been changed accordingly in the revised manuscript (Page 2, Line 73-75).

Comment 8: Line 83: The legs were used immediately after the animal was slaughtered, or the carcasses were stored for 24 h to take the piece later.

Response 8: Thank you for your careful review and comment. Actually, the pig hind legs were kept at 4 °C for 48 h after slaughter until use. According to your valuable comment, this information has been added in the revised manuscript (Page 2, Line 84-86).

Comment 9: Line 90: Idem

Response 9: The same to Response 5.

Comment 10: Line 104: Idem

Response 10: The same to Response 5. 

Comment 11: Line 105: Briefly, samples.

Response 11: Thank you for your careful review. It has been corrected in the revised manuscript (Page 3, Line 105).

Comment 12: Line 105: insert mixing conditions.

Response 12: Thank you for your valuable suggestion. Actually, approximately 20 mg sample was used for the mixing and further analysis. According to your suggestion, this condition has been added in the revised manuscript (Page 3, Line 105-106).

Comment 13: Line 106: insert equipment information (model, brand, country).

Response 13: Thank you for your careful review and valuable comments. The vortex equipment was bought from the Scilogex company in USA, and its model was MX-S. Accordingly, the information has been added in the revised manuscript (Page 3, Line 106-107).

Comment 14: Line 107: insert equipment information (model, brand, country).

Response 14: Thank you for your careful review and valuable comments. The ultrasonic water bath was purchased from the Kun Shan Ultrasonic Instruments Co., Ltd. (Jiangsu, China), and its model was KQ5200E. Accordingly, the information has been added in the revised manuscript (Page 3, Line 107-108).

Comment 15: Line 108: 14,000.

Response 15: Thank you for your careful review. The “14000” has been modified to “14,000” accordingly in the revised manuscript (Page 3, Line 109).

Comment 16: Line 111: 14,000.

Response 16: Thank you for your careful review. The “14000” has been modified to “14,000” accordingly in the revised manuscript (Page 3, Line 112).

Comment 17: Line 113: Analysis.

Response 17: Thank you for your careful review. It has been corrected accordingly in the revised manuscript (Page 3, Line 115).

Comment 18: Line 114: Idem

Response 18: The same to Response 5.

Comment 19: Line 120: °C.

Response 19: Thank you for your careful review. It has been corrected accordingly in the revised manuscript (Page 3, Line 122).

Comment 20: Line 179: 83.28%.

Response 20: Thank you for your careful review. It has been corrected accordingly in the revised manuscript (Page 4, Line 184).

Comment 21: Line 219: acids and derivatives.

Response 21: Thank you for your careful review. The “carboxylic acids & derivatives” has been changed to “carboxylic acids and derivatives” accordingly in the revised manuscript (Page 6, Line 226).

Comment 22: Line 220: lipids and lipid-like.

Response 22: Thank you for your careful review. The “lipids & lipid-like” has been changed to “lipids and lipid-like” accordingly in the revised manuscript (Page 6, Line 227).

Comment 23: Line 222: acids and derivatives.

Response 23: Thank you for your careful review. The “carboxylic acids & derivatives” has been changed to “carboxylic acids and derivatives” accordingly in the revised manuscript (Page 6, Line 229).

Comment 24: Line 223,224,226: use “and” instead of “&”.

Response 24: Thank you for your careful review. The “&” has been changed to “and” accordingly in the revised manuscript (Page 6, Line 230-234).

Comment 25: Line 230: Idem

Response 25: The same to Response 5. 

Comment 26: Line 232: Idem

Response 26: The same to Response 5.

Comment 27: Line 241, 246, 248, 284, 309: Idem

Response 27: The same to Response 5. 

Comment 28: Line 312: The resolution of the figure needs to be improved; when zooming in, the letter becomes distorted and it is difficult to read the text.

Response 28: Thank you for your careful review and valuable suggestion. According to your valuable suggestion, Figure 6 has been updated to a new figure with higher resolution in the revised manuscript (Page 10, Figure 6). 

Comment 29: use “and” instead of “&” (make change through document text).

Response 29: Thank you for your valuable suggestion. According to your suggestion, all the “&” has been changed to “and” across the whole manuscript in the revised manuscript (Page 7, Line 260; Page 11, Line 327-329; Page 11, Line 363; Page 12, Line 394).

Comment 30: Line 336: Idem

Response 30: The same to Response 5. 

Comment 31: Line 351: remove text space… [27,30].

Response 31Thank you for your careful review. The text space has been deleted accordingly in the revised manuscript (page 11, line 357).

Comment 32:Lines 364, 370: Same as above

Response 32Same as response 5.

Round 2

Reviewer 1 Report

Comments and Suggestions for Authors

The authors have improved the manuscript according to the comments of both reviewers. I still have some remaining comments, which are listed below: 

·         Comment & Response 7: Although I appreciate the addition on p. 4, it would be more interesting to read why – to which specific purpose - PCA and OPLS-DA were used in this study and this paper specifically, instead of the general theory.

·         Comment & Response 8: I do not find the added Figure S1 very informative. The TICs do not allow assessment of signal drift. This may be done by modelling and plotting the QCs using PCA, in the PCA score plot. The QCs should cluster closely together.  

·         Comment & Response 10: Please add a list of the in-house standards used.

Author Response

Comment 1: Although I appreciate the addition on p. 4, it would be more interesting to read why – to which specific purpose - PCA and OPLS-DA were used in this study and this paper specifically, instead of the general theory.

Response 1: Thank you for your valuable suggestion. Actually, PCA was used for the overall metabolite profile analysis of FH and LH groups, including both inter-group and intra-group variations, as it’s an unsupervised learning method. OPLS-DA was used mainly for the metabolite differentiation analysis between FH and LH, especially for the DMs screening based on VIP values in the further analysis, since it’s a supervised learning method with more focuses on inter-group variations. According to your suggestion, modifications have been made in the revised manuscript (Page 4, Lines 176-180).

Comment 2: I do not find the added Figure S1 very informative. The TICs do not allow assessment of signal drift. This may be done by modelling and plotting the QCs using PCA, in the PCA score plot. The QCs should cluster closely together.

Response 2: Thanks for your valuable suggestion. According to your suggestion, Figure S1 has been changed to the principal component analysis (PCA) plots of metabolites from quality control (QC) and test samples (FH and LH) at both ESI+ (a) and ESI- (b) in the Supplementary Materials of the revised manuscript (Page 3, Lines 126-128; Page 12, Lines 411-412).

Comment 3: Please add a list of the in-house standards used.

Response 3: Thank you for your valuable comment. The in-house database with available authentic standards was provided by Shanghai Applied Protein Technology Co., Ltd., which was also been used by many peer-reviewed publications for the metabolite identification, such as Gu et al. [1] and Luo et al. [2]. Moreover, many parameters, such as mass matching (<25 ppm), retention times, secondary spectra matching, and collision cross-section values, were considered during the metabolite identification to make sure the compound annotation confidence reach to a level 1 standard, as discussed by the Compound Identification work group of the Metabolomics Society at the 2017 annual meeting of the Metabolomics Society [3]. The above information has been added accordingly in the revised manuscript (Page 3, Lines 145-148). Meanwhile, according to your comment, the identification results of all 450 defined metabolites from FH and LH through UPLC-MS/MS at both ESI+ and ESI-, including the metabolite ID, official name, adduct in MS/MS spectra, m/z value, and retention time, have been added as Table S1 in the Supplementary Materials of the revised manuscript (Page 6, Lines 219-220; Page 13, Lines 412-413).

References:

[1] Gu, Z., Li, L., Tang, S., Liu, C., Fu, X., Shi, Z., & Mao, H. (2018). Metabolomics reveals that crossbred dairy buffaloes are more thermotolerant than Holstein cows under chronic heat stress. Journal of Agricultural and Food Chemistry, 66(49), 12889-12897.

[2] Luo, D., Deng, T., Yuan, W., Deng, H., & Jin, M. (2017). Plasma metabolomic study in Chinese patients with wet age-related macular degeneration. BMC ophthalmology, 17, 1-9.

[3] Blaženović, I., Kind, T., Ji, J., & Fiehn, O. (2018). Software tools and approaches for compound identification of LC-MS/MS data in metabolomics. Metabolites, 8(2), 31.

Round 3

Reviewer 1 Report

Comments and Suggestions for Authors

The authors have revised the paper accordingly.

Comments on the Quality of English Language

The English could be improved in the newly added/altered parts.